# Aotaphenazine, a rare hydrophenazine, targets topoisomerase II with anticancer efficacy: *In silico* to *in vitro* evidence

Ahmed M. Metwaly[1]*, Ibrahim H. Eissa[2]*, Wael M. Afifi[1,3], Eslam B. Elkaeed[4], Aisha A. Alsfouk[5], Ibrahim M. Ibrahim[6], Mohamed S. Abdelfattah[7]

1 Pharmacognosy and Medicinal Plants Department, Faculty of Pharmacy (Boys), Al-Azhar University, Cairo, Egypt, 2 Pharmaceutical Medicinal Chemistry & Drug Design Department, Faculty of Pharmacy (Boys), Al-Azhar University, Cairo, Egypt, 3 Department of Pharmacognosy, Faculty of Pharmacy, Sinai University—Kantara Branch, Ismailia, Egypt, 4 Department of Pharmaceutical Sciences, College of Pharmacy, AlMaarefa University, Riyadh, Saudi Arabia, 5 Department of Pharmaceutical Sciences, College of Pharmacy, Princess Nourah bint Abdulrahman University, Riyadh, Saudi Arabia, 6 Biophysics Department, Faculty of Science, Cairo University, Giza, Egypt, 7 Chemistry Department, Faculty of Science, Helwan University, Cairo, Egypt

* ametwaly@azhar.edu.eg (AHM); Ibrahimeissa@azhar.edu.eg (IHE)

## Abstract

### Background

Topoisomerase II (Topo II) remains a validated target for anticancer therapy, with many clinically used agents acting via DNA intercalation and enzyme inhibition. However, their clinical use is limited by severe toxicity and resistance. In this study, we investigate aotaphenazine, a rare hydrophenazine derivative isolated from *Streptomyces* sp. IFM 11694 as a potential novel Topo II inhibitor with selective anticancer activity.

### Methods

Molecular docking and 400 ns molecular dynamics (MD) simulations were employed to evaluate aotaphenazine's binding mode within the DNA–Topo II complex (PDB: 3QX3). MM-GBSA calculations quantified interaction energetics, while ProLIF and PLIP analyses detailed the interaction patterns. Topo II inhibition was assessed via *in vitro* enzymatic assays. Cytotoxicity (MTT) assays were conducted against a panel of human cancer and normal cell lines. Flow cytometry was used to evaluate apoptosis and cell cycle progression in MDA-MB-231 cells.

### Results

aotaphenazine demonstrated a docking binding energy of –19.12 kcal/mol and remained stably intercalated within the DNA groove during MD simulations. MM-GBSA analysis showed a total binding free energy of –29.81 kcal/mol, driven

**Data availability statement:** All relevant data are within the manuscript and its Supporting Information files.

**Funding:** This research was funded by Princess Nourah bint Abdulrahman University Researchers Supporting Project number (PNURSP2025R116), Princess Nourah bint Abdulrahman University, Riyadh, Saudi Arabia The funders had no role in study design, data collection and analysis, decision to publish, or preparation of the manuscript.

**Competing interests:** The authors have declared that no competing interests exist.

primarily by van der Waals forces. Interaction profiling identified consistent π-stacking with Cyt8 and Thy9, and strong binding contributions from Ade12 and Gua13. Enzymatic assays confirmed Topo II inhibition with an $IC_{50}$ of 45.01 nM, comparable to doxorubicin (30.16 nM). *In vitro* cytotoxicity analysis revealed moderate activity across cancer cell lines ($IC_{50} = 26.30–54.35$ μM) and significantly reduced toxicity in normal WI-38 and WISH cells ($IC_{50} = 69.86$ μM and 84.72 μM, respectively). Flow cytometry showed that aotaphenazine induced early (20.98%) and late apoptosis (42.80%), along with S-phase cell cycle arrest (43.99%) and a marked reduction in the G2/M population in MDA-MB-231 cells.

## Conclusion

aotaphenazine exhibits a compelling combination of Topo II inhibition, DNA intercalation, and selective anticancer activity, supported by both computational modeling and biological validation. Its lower cytotoxicity toward normal cells and ability to induce apoptosis and cell cycle arrest suggest strong therapeutic potential. These findings establish aotaphenazine as a promising lead compound for the development of safer and more selective Topo II-targeting anticancer agents.

## 1. Introduction

Cancer remains a leading cause of mortality worldwide, characterized by uncontrolled cell proliferation, genetic instability, and resistance to apoptosis [1]. Targeting DNA topology and replication process has emerged as one of the most successful therapeutic strategies in oncology [2]. Among these, DNA topoisomerase II (Topo II) is a well-established molecular target due to its critical role in maintaining DNA topology during replication, transcription, and chromosomal segregation [3]. The enzyme introduces transient double-stranded breaks to relieve superhelical tension and is essential for cell viability, especially in rapidly dividing cancer cells [4]. Consequently, Topo II inhibitors have become cornerstone agents in the chemotherapeutic arsenal, with drugs such as doxorubicin, etoposide, and mitoxantrone widely used across a range of malignancies [5,6].

Despite their clinical efficacy, current Topo II-targeting agents suffer from several critical limitations, including dose-limiting toxicities, such as cardiotoxicity in the case of anthracyclines [7], genotoxic side effects [8], and the emergence of resistance [9,10]. These drawbacks have spurred interest in the discovery of novel Topo II inhibitors with improved selectivity, reduced toxicity, and novel chemical scaffolds. Additionally, the development of cancer-selective cytotoxic drugs capable of preferentially targeting malignant over normal cells remains a high priority in anticancer drug discovery.

Natural products continue to play a pivotal role in anticancer drug development, providing structurally diverse, biologically active compounds with unique mechanisms of action [11]. In particular, phenazine derivatives, a class of coloured

nitrogen-containing heterocycles commonly produced by *Streptomyces* and *Pseudomonas* species, have shown broad-spectrum antimicrobial and anticancer properties [12–14]. Several synthetic and semi-synthetic phenazines have been explored for their ability to intercalate into DNA and inhibit topoisomerase enzymes, making them attractive scaffolds for further development [15].

The integration of *in silico* and *in vitro* methods proves particularly advantageous for initial screening, efficiently narrowing down potential candidates before resource-intensive experimental validation [16]. Our research group has successfully implemented integrated *in silico-in vitro* strategies across multiple therapeutic areas, encompassing SARS-CoV-2 antiviral development [17–19], microbial virulence targeting [20,21], anticancer therapeutics [22–25], and bioactive compound delivery optimization [26–28].

In this context, aotaphenazine, (Fig 1), a rare hydrophenazine derivative isolated from *Streptomyces* sp. IFM 11694 collected from Namegata, Ibaraki, Japan, represents a novel addition to the phenazine family [29,30]. From a medicinal chemistry perspective, preliminary structural characterization indicates that aotaphenazine features a planar aromatic core – a key structural motif shared with known intercalative agents that enables both DNA intercalation and Topo II inhibition. However, unlike many classical phenazine derivatives, aotaphenazine also exhibits structural simplicity and modifiable side chains, making it amenable to synthetic optimization. Despite its rarity and limited prior investigation, aotaphenazine presents as a promising candidate for targeted anticancer therapy based on its natural origin, manageable physicochemical profile, and likely mechanism of action.

In this study, we provide a comprehensive *in silico* and *in vitro* evaluation of aotaphenazine as a novel Topo II inhibitor and anticancer agent. We employed computational approaches including molecular docking, molecular dynamics (MD) simulations for 400 ns, MM-GBSA free energy analysis, ProLIF and PLIP to characterize its interaction with the DNA–Topo II complex. These findings were complemented by *in vitro* biochemical assays assessing Topo II inhibition potency and cytotoxicity across a panel of human cancer and normal cell lines. Additionally, flow cytometry-based assays were conducted to assess its effects on apoptosis induction and cell cycle progression, particularly in the triple-negative breast cancer cell line MDA-MB-231. Our findings provide compelling evidence that aotaphenazine is a viable lead compound with dual intercalative and Topo II-inhibitory action, offering selective cytotoxicity and mechanistic promise for further anticancer drug development.

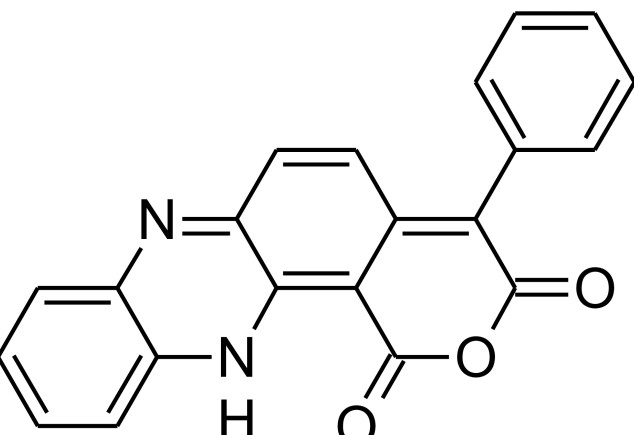

**Fig 1. Aotaphenazine's chemical structure.**

## 2. Results and discussions

### 2.1. Computational studies

**2.1.1. Molecular docking.** The molecular docking study of aotaphenazine presents promising insights into its potential as a lead compound for topoisomerase II (Topo II) inhibition. With a binding free energy ($\Delta G$) of −19.12 kcal/mol against the DNA–Topo II complex (PDB ID: 3QX3), aotaphenazine demonstrates a spontaneous and thermodynamically favorable interaction. Although this value is modestly lower than that of doxorubicin (−27.39 kcal/mol), it remains within a biologically significant range. This suggests that aotaphenazine has the essential structural and energetic attributes necessary to serve as a foundational scaffold for anticancer drug development.

One of the most striking features of aotaphenazine is its planar phenazine core, which closely mirrors the flat, aromatic system of doxorubicin. This planarity is critical for intercalative binding, as it allows the molecule to effectively slip between DNA base pairs, facilitating π-π stacking interactions. Both aotaphenazine and doxorubicin form hydrophobic contacts with key nucleobases such as Guanine 13 (Gua13), Adenine 12 (Ade12), and Thymine 9 (Thy9). These shared interaction points suggest a similar DNA intercalation mode, anchoring the ligands in the major groove of the double helix. Moreover, the presence of multiple hydrophobic interactions in both compounds contributes significantly to the stabilization of the DNA–ligand complex (Fig 2).

In addition to the overlapping base contacts, aotaphenazine also interacts with Arginine 503 (Arg503), a residue involved in the protein portion of the DNA–Topo II complex. This is conceptually parallel to doxorubicin's engagement with Lysine 456 (Lys456), suggesting that both molecules extend their interactions beyond DNA intercalation into enzyme anchoring. Although doxorubicin forms a more extensive hydrogen bonding network due to its sugar moiety and polar groups, this structural difference highlights an opportunity for future optimization of aotaphenazine. Specifically, chemical modification strategies can focus on introducing hydrophilic substituents to promote similar H-bonding, thereby enhancing binding affinity and specificity. The docking results, combined with prior biological evidence of activity against TRAIL-resistant cancer cells, position aotaphenazine as more than an academic curiosity—it is a viable chemotherapeutic proto-type with room for rational enhancement.

In conclusion, the binding profile of aotaphenazine reveals notable parallels with doxorubicin, particularly in DNA base interactions and the intercalative mode of action. Although it currently lacks the extensive hydrogen bonding network of the latter, its simpler structure offers clear paths for medicinal chemistry optimization. With strategic modifications to enhance hydrogen bonding and enzyme engagement, aotaphenazine holds the potential to evolve into a powerful and selective Topo II inhibitor, addressing both drug resistance and efficacy challenges in cancer therapy.

**2.1.2. Molecular dynamics (MD) simulation and binding interaction analysis.** The MD simulations (400 ns) provided critical insight into the temporal stability and binding behavior of aotaphenazine within the DNA–Topoisomerase II complex. Analysis of the protein's root mean square deviation (RMSD) revealed a moderate initial increase during the early phases of the simulation, consistent with structural relaxation, followed by stabilization around 5 Å, indicating no significant conformational disruption and suggesting that the complex remained globally stable throughout the 400 ns trajectory (Fig 3A). A similar trend was observed for the ligand RMSD, which initially increased during the first 100 ns, possibly reflecting ligand adaptation or reorientation within the intercalation site, before decreasing transiently to 3.5 Å and then stabilizing around 4.5 Å (Fig 3B). This stabilization phase suggests that aotaphenazine achieved a stable and energetically favorable binding conformation in the second half of the simulation.

One of the most notable dynamic changes was observed in the hydrogen bonding pattern. While the first half of the simulation showed minimal hydrogen bonding, the second half demonstrated the formation of a stable average of two hydrogen bonds between the ligand and the macromolecule (Fig 3C). This shift indicates a reorganization of polar functional groups within aotaphenazine, likely reflecting its accommodation into a more optimal orientation to engage with specific hydrogen bond donors or acceptors in the binding pocket. This evolution supports the hypothesis of a progressive

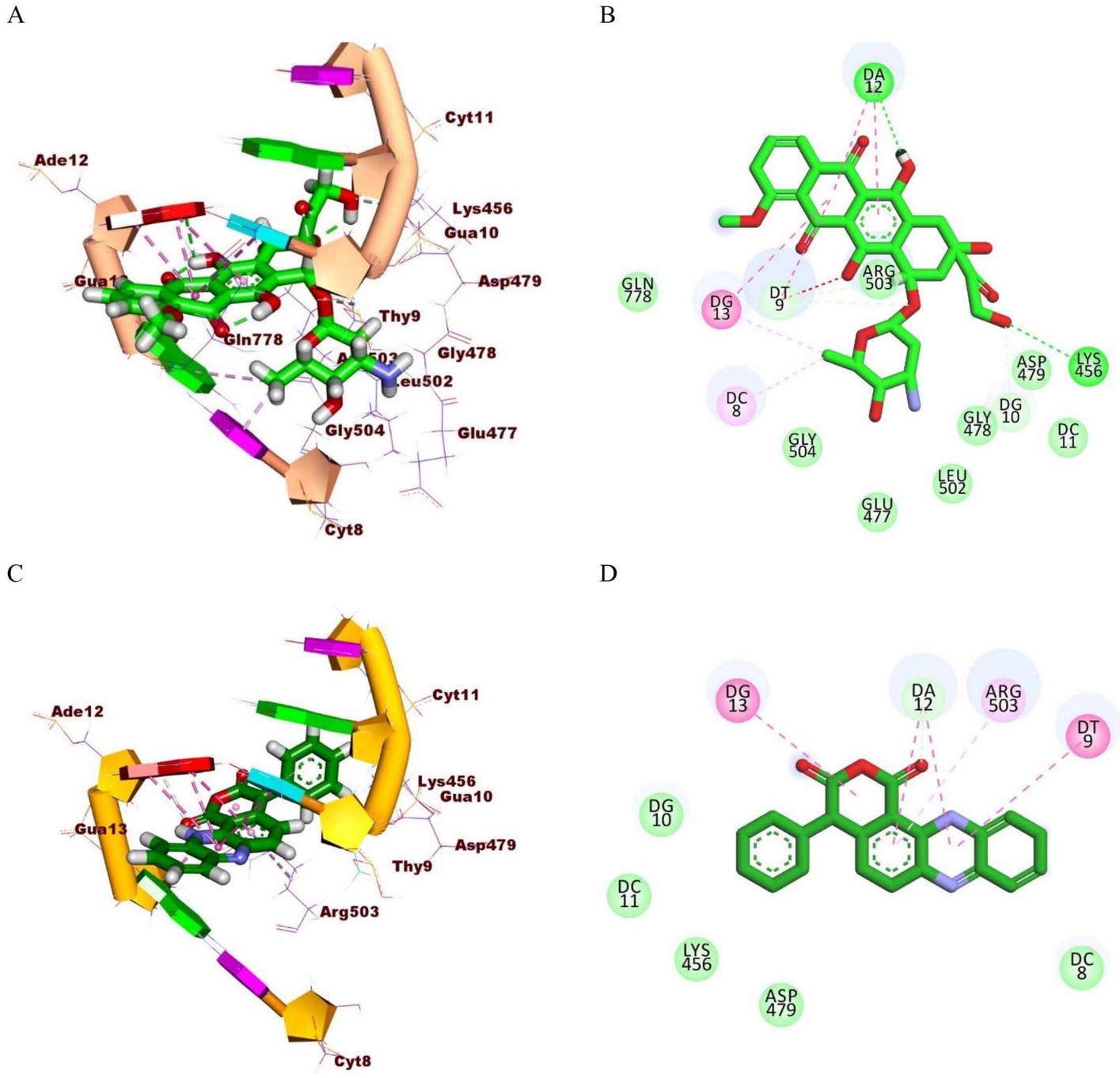

**Fig 2. Molecular interactions of doxorubicin and aotaphenazine within the DNA–Topoisomerase II (Topo II) complex.** (A, B) Three-dimensional (A) and two-dimensional (B) interaction profiles of doxorubicin bound to the DNA–Topo II complex, showing its characteristic intercalation between DNA base pairs. (C, D) Three-dimensional (C) and two-dimensional (D) interaction views of aotaphenazine within the same complex, illustrating its stable DNA intercalation and key binding interactions. Hydrogen bonds are depicted as green dashed lines, and π–π stacking or hydrophobic interactions as pink dashed lines. The models were generated from molecular docking using the crystal structure of the DNA–Topo II complex (PDB ID: 3QX3).

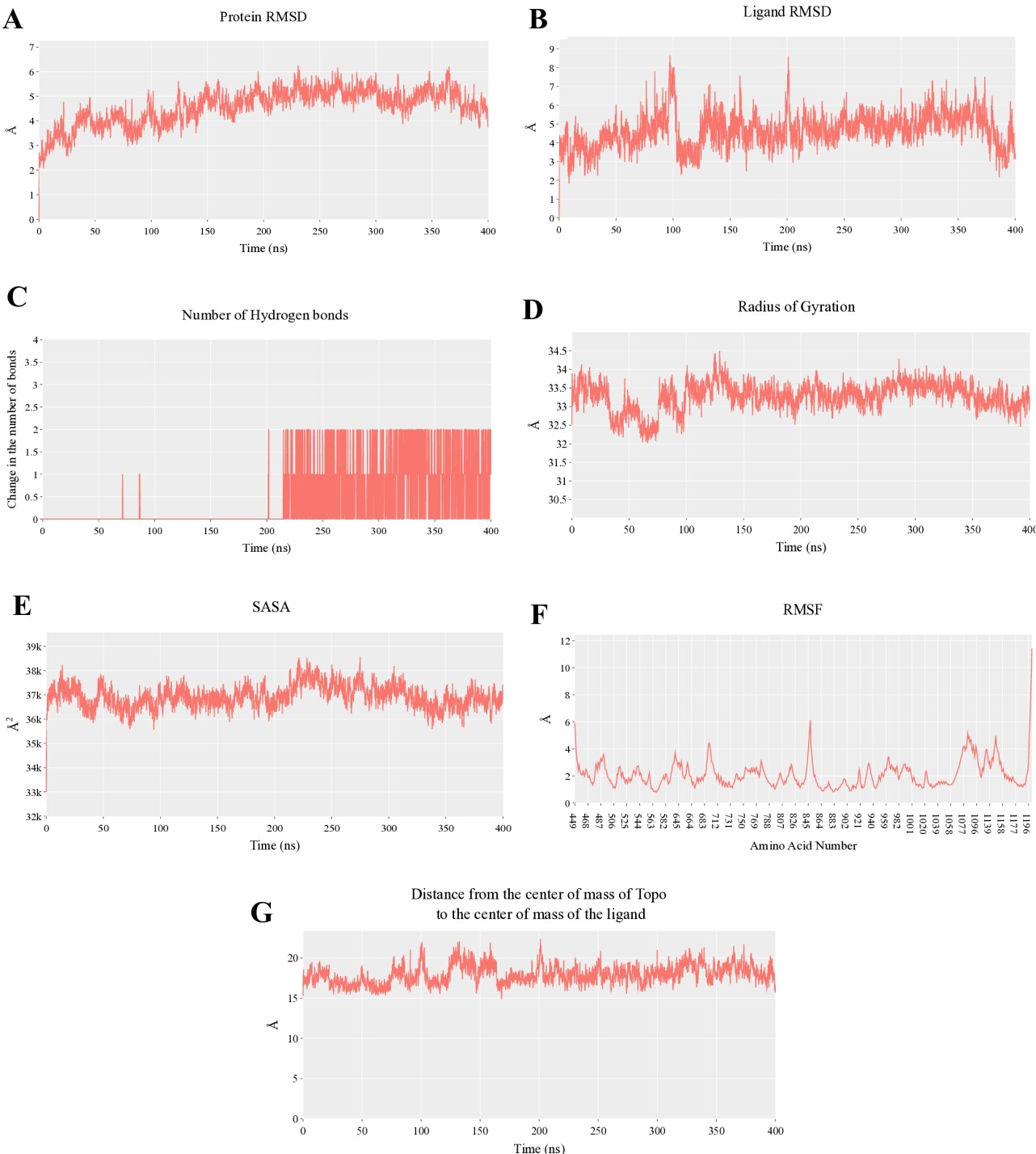

**Fig 3. Molecular dynamics (MD) trajectories of the aotaphenazine–DNA–Topoisomerase II (Topo II) complex over a 400 ns simulation time.** (A) RMSD of the protein backbone, indicating overall structural stability of the Topo II–DNA complex. (B) RMSD of aotaphenazine, reflecting the conformational stability within the binding pocket. (C) Hydrogen bond count between aotaphenazine and the complex throughout the 400 ns, demonstrating interaction persistence. (D) RoG of the protein, showing compactness and structural integrity. (E) SASA, representing protein exposure to solvent. (F) RMSF of residues, highlighting local flexibility across the protein structure. (G) Distance between the centers of mass of aotaphenazine and Topo II, showing the stability of ligand binding and intercalation over the 400 ns.

conformational fitting of the ligand, characteristic of intercalators that stabilize upon achieving ideal base stacking and groove orientation.

Furthermore, the Radius of Gyration (RoG) and Solvent Accessible Surface Area (SASA) remained remarkably constant throughout the simulation (averaging 33.2 Å and 37,000 Å², respectively; Fig 3D and 3E). This consistency underscores the structural compactness and solvent exposure stability of the protein-ligand complex, reinforcing the observed conformational integrity. Minimal root mean square fluctuations (RMSF) in most Cα atoms (Fig 3F), with only occasional spikes—likely corresponding to terminal or loop regions—further confirm that ligand binding did not induce undue structural perturbation to the protein scaffold. Additionally, the center-of-mass distance between the ligand and the protein remained stable at approximately 17 Å (Fig 3G), affirming that aotaphenazine remained consistently associated with the complex, supporting the concept of a retentive and functionally stable binding mode.

**2.1.3. MM-GBSA studies.** Energetic analysis using MM-GBSA further substantiated the binding strength of aotaphenazine. The average total binding free energy was calculated at –29.81 kcal/mol, denoting a favorable and spontaneous binding event (Fig 4). The decomposition of this energy profile highlighted a predominant contribution from van der Waals interactions (–40.16 kcal/mol), with modest electrostatic contributions (–5.54 kcal/mol). These findings emphasize the hydrophobic nature of aotaphenazine's binding, consistent with its planar phenazine scaffold, which is known to favor π-stacking and intercalation into nucleic acid base pairs. The relatively lower electrostatic contribution is also in line with the limited presence of strongly polar or charged functional groups, which were nonetheless capable of forming a stable hydrogen bond network in the latter half of the simulation.

Per-residue energy decomposition identified the nucleic acid bases as the primary binding partners, confirming aotaphenazine's preference for the DNA interface rather than the protein surface (Fig 5). Key bases within the binding region—Guanine 13 (–3.69 kcal/mol), Adenine 12 (–3.46 kcal/mol), Cytosine 8 (–2.54 kcal/mol), and Thymine 9 (–1.78 kcal/mol)—contributed significantly to the binding energy. These bases correspond to classical intercalation zones and reinforce the proposed intercalative mechanism of action. Interestingly, the protein residue Arginine 503 also displayed a significant energetic contribution (–2.76 kcal/mol), indicating that aotaphenazine may engage in π–cation or side-chain polar interactions with residues near the DNA–enzyme interface, thereby stabilizing its anchoring and enhancing specificity.

**2.1.4. Protein–ligand interaction fingerprint (ProLIF) analysis.** ProLIF interaction profiling provided detailed insight into the nature and persistence of non-covalent interactions formed between aotaphenazine and the DNA–Topo II complex throughout the simulation trajectory (Fig 6). By evaluating the frequency and type of contacts frame-by-frame, this analysis offers a comprehensive fingerprint of the binding interactions, further substantiating the ligand's intercalative binding mode observed in docking and MD simulations.

Aotaphenazine exhibited highly persistent hydrophobic interactions with Thymine 9, present in 95% of all simulation frames, and with Cytosine 8 in 83.1%, suggesting that the ligand remains deeply embedded within a non-polar pocket formed by adjacent DNA bases. These hydrophobic contacts are crucial for maintaining the structural integrity of aotaphenazine, where van der Waals interactions dominate the energetic landscape, as previously confirmed by MM-GBSA calculations. The presence of such stable, nonpolar interactions with multiple consecutive bases implies that aotaphenazine is not merely surface-bound or groove-associated, but is indeed positioned within a well-defined intercalation site, mimicking the behavior of classical anthracyclines and phenazine-based intercalators.

Furthermore, π–π stacking interactions between aotaphenazine and Cytosine 8 were observed in 71% of the trajectory, indicating a strong planar alignment of the phenazine core with the aromatic ring system of the DNA base pair. This is particularly significant, as π-stacking not only contributes to the ligand's thermodynamic stability within the DNA duplex but also enhances specificity by favoring geometrically compatible intercalation zones. These interactions are facilitated by the rigid and electron-rich nature of aotaphenazine's tricyclic scaffold, which aligns favorably with the DNA base π-clouds, allowing for efficient electron delocalization and dispersion-based stabilization.

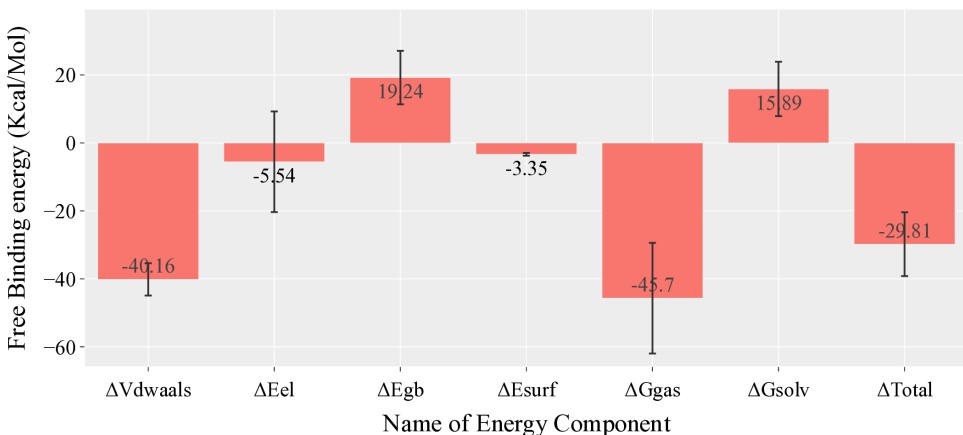

**Fig 4. Energy decomposition profile from MM-GBSA analysis of the aotaphenazine–DNA–Topoisomerase II (Topo II) complex.** The plot illustrates the various energy components contributing to the total ΔG_bind, including van der Waals, electrostatic, polar solvation, and SASA energies.

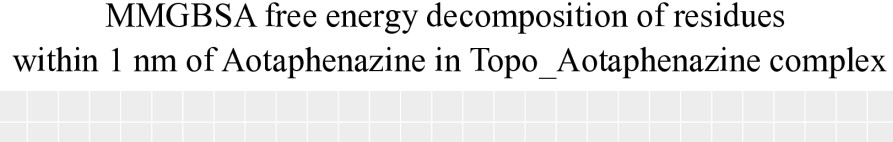

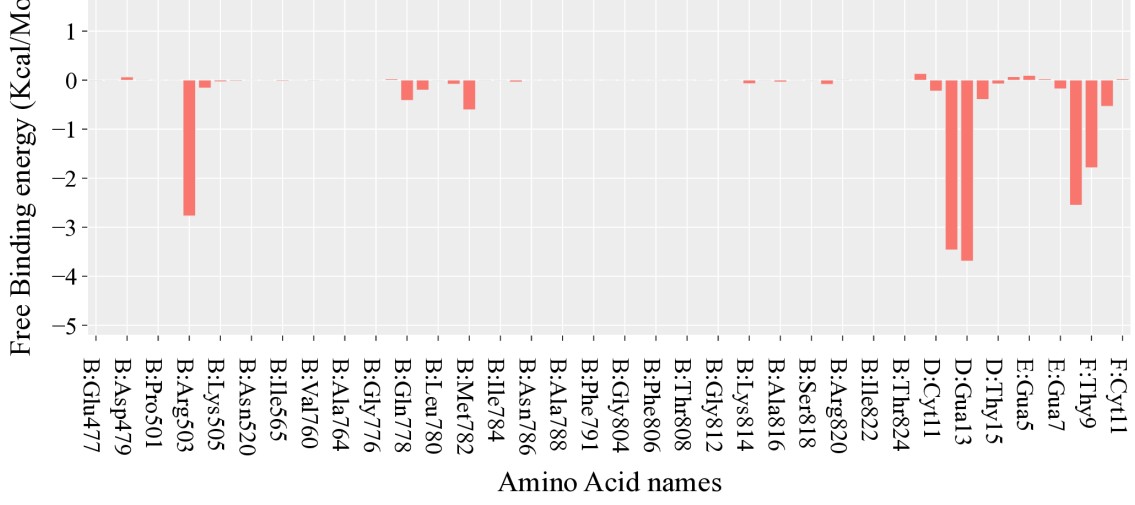

**Fig 5. MM-GBSA free energy decomposition analysis of aotaphenazine-DNA–Topo II Complex.**

The spatial persistence and high frequency of both hydrophobic and π-stacking interactions strongly support aotaphenazine's ability to achieve a well-defined, energetically favorable intercalative orientation. This mode of binding is consistent with its observed preference for DNA base interactions over protein residues and aligns with the dominant contributions from Cytosine 8, Thymine 9, Adenine 12, and Guanine 13 identified in the residue decomposition analysis. Together, these findings illustrate how aotaphenazine leverages specific and sustained molecular interactions to remain stably intercalated, contributing to its ability to disrupt DNA processing by Topo II and induce downstream cytotoxic effects.

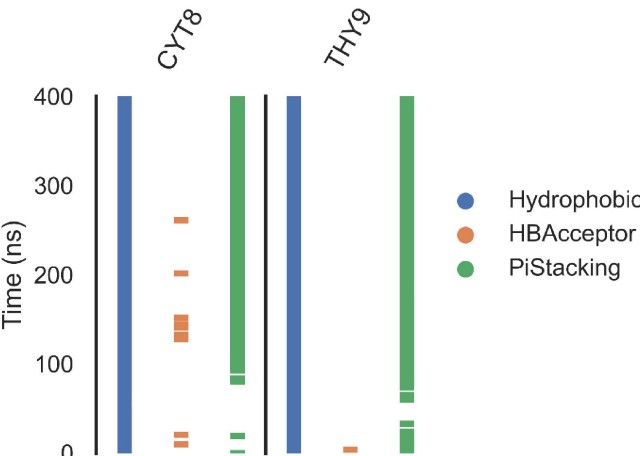

**Fig 6. ProLIF interaction fingerprint analysis of the aotaphenazine–DNA–Topoisomerase II (Topo II) complex.**

**2.1.5. Protein–ligand interaction profiler (PLIP) studies.** Finally, representative three-dimensional binding poses were extracted through trajectory clustering and analyzed using the PLIP, with visualizations rendered via PyMol (Fig 7). These representative frames were selected from the most populated structural clusters to capture the most stable and frequently occurring conformational states of the aotaphenazine–Topo II–DNA complex during the simulation. Such clustering ensures that transient, non-representative interactions are filtered out, allowing a more accurate depiction of the ligand's most probable binding mode.

The structural visualization confirmed that aotaphenazine consistently adopts an intercalative pose, with its rigid, planar phenazine core oriented in parallel alignment with the DNA base pairs. This geometry is indicative of π–π stacking and is a hallmark of classical intercalators. The molecule was found nestled between base pairs within the DNA duplex, with clear evidence of base displacement and ring stacking, validating the interpretation from both MM-GBSA and ProLIF analyses. The molecular surface and electrostatic potential mapping further demonstrated the complementarity between the ligand and the DNA groove, supporting its ability to occupy a favorable intercalation site with minimal steric or electrostatic clash.

Importantly, hydrogen bond networks observed in the latter half of the simulation were structurally captured in these representative frames. PLIP analysis revealed the formation of up to two stable hydrogen bonds, involving interactions between polar moieties on aotaphenazine and nearby nucleotide bases or side chains such as Arg503, which had also shown significant energetic contributions in residue decomposition analyses. These hydrogen bonds were not present in the early simulation snapshots, highlighting a time-dependent reorientation of functional groups that reflects the dynamic optimization of the binding conformation. This observation reinforces the idea that aotaphenazine transitions from an initial hydrophobic docking phase into a stabilized, intercalated state enriched with both van der Waals and hydrogen bonding contributions.

Moreover, the static frames provided critical visual confirmation of binding pocket depth, orientation of key functional groups, and the extent of groove penetration, all of which align with known characteristics of DNA-targeted intercalative ligands. The observed ligand conformation and proximity to DNA bases—including Cyt8, Thy9, Ade12, and Gua13—were in strong agreement with the residues identified through MM-GBSA decomposition, offering multi-level validation of the interaction map.

In summary, this final structural analysis not only corroborated the trajectory-based findings but also offered direct atomic-level insight into the binding orientation, interaction specificity, and adaptability of aotaphenazine. These structural

**C1**

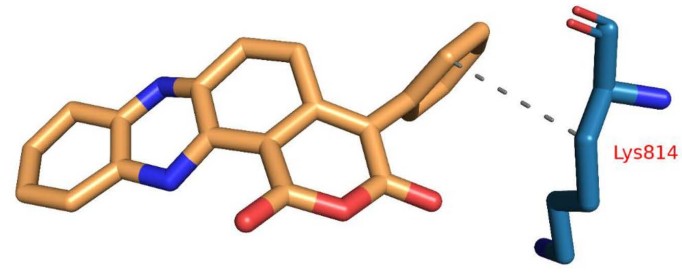

**C2**

**C3**

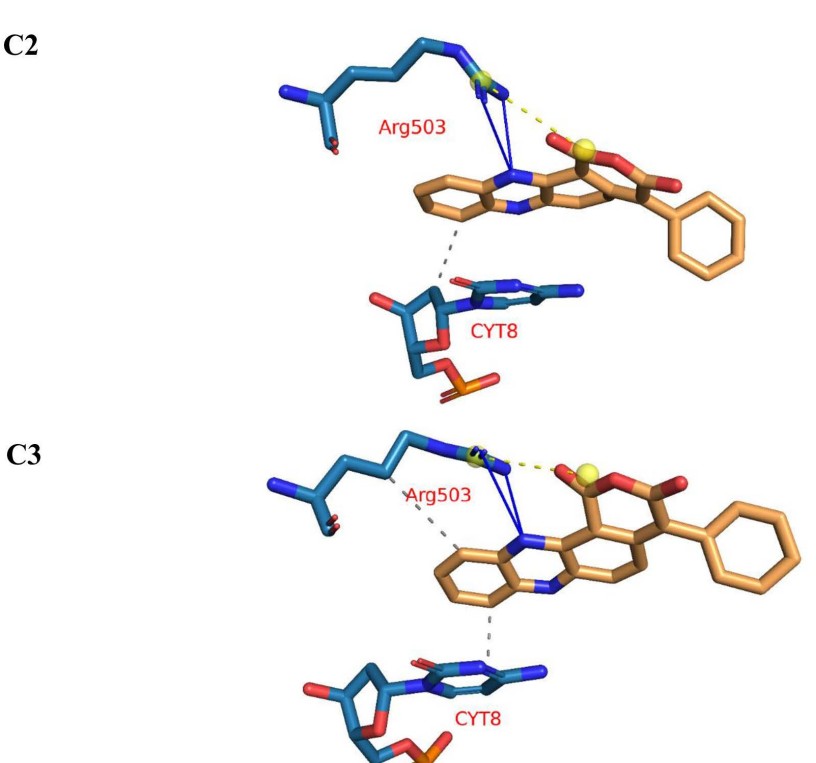

**Fig 7. PLIP interaction analysis of the aotaphenazine–DNA–Topoisomerase II (Topo II) complex showing representative three-dimensional binding of three poses (C1, C2, and C3).**

frames underscore the potential of aotaphenazine to serve as a mechanistically validated DNA intercalator and Topo II inhibitor, strengthening its candidacy as a lead scaffold for further anticancer drug development.

Together, these MD simulation results offer strong mechanistic insight into the favorable, stable, and selective binding behavior of aotaphenazine. The data reveal a compound that transitions from loose initial engagement to a tightly intercalated and hydrogen-bonded conformation, capable of disrupting the DNA–Topo II interface. This binding profile, dominated by hydrophobic and base-specific π-stacking, alongside supportive polar contacts, underscores aotaphenazine's potential

as a dual-functioning anticancer agent—capable of Topo II inhibition and DNA intercalation, with a distinct interaction footprint compared to traditional anthracyclines.

## 2.2. *In vitro* evaluations

**2.2.1. Topo II inhibition.** The biological evaluation of aotaphenazine for Topo II inhibition revealed an $IC_{50}$ value of 45.01 nM, indicating a potent inhibitory effect against the enzyme. While this value is slightly higher than that of the reference compound doxorubicin ($IC_{50} = 30.16$ nM), it nonetheless reflects a comparable level of enzymatic inhibition. Given that doxorubicin is a well-established clinical Topo II inhibitor, these results suggest that aotaphenazine exhibits strong and therapeutically relevant activity *in vitro*.

The marginal difference in potency ($\Delta IC_{50} \approx 15$ nM) may be attributed to differences in molecular size, hydrophilicity, and hydrogen bonding potential. As previously described in the molecular docking and MD simulation results, doxorubicin forms multiple hydrogen bonds and electrostatic interactions due to its sugar moiety and polar substituents, which enhance its binding affinity and inhibition potency. In contrast, aotaphenazine relies more heavily on π–π stacking and hydrophobic interactions with DNA bases, as evidenced by the high-frequency interactions with Cytosine 8 and Thymine 9 in the ProLIF analysis, and the strong van der Waals energy contributions revealed by MM-GBSA calculations.

Despite lacking the extensive hydrogen bonding network of doxorubicin, aotaphenazine's $IC_{50}$ value demonstrates that its intercalative binding mode is sufficient to disrupt Topo II function effectively. This finding is particularly encouraging considering that aotaphenazine represents a simpler chemical scaffold, which may allow greater synthetic flexibility and fewer off-target effects than anthracycline-based inhibitors. Furthermore, its phenazine backbone offers multiple sites for structural optimization, including the potential to enhance aqueous solubility, target selectivity, or improve pharmacokinetics.

Taken together, these data establish aotaphenazine as a highly promising lead compound for Topo II inhibition, with potency in the nanomolar range and mechanistic support from computational modeling. Future efforts may focus on optimizing the ligand through rational design strategies that incorporate polar functional groups or sugar mimetics to further lower the $IC_{50}$ while maintaining the favorable intercalative characteristics observed in this study.

**2.2.2. In vitro cytotoxicity evaluation.** The *in vitro* cytotoxic profile of aotaphenazine was assessed across a panel of human cancer cell lines representing diverse tissue origins (Table 1), including liver (HePG-2), breast (MCF-7, MDA-MB-231), colon (HCT-116, Caco-2), prostate (PC-3), and cervix (HeLa), alongside two normal cell lines—WI-38 (lung fibroblasts) and WISH (amniotic epithelial cells). The $IC_{50}$ values obtained were compared with those of doxorubicin (DOX), a clinically used Topo II inhibitor, to evaluate relative potency and selectivity.

Aotaphenazine exhibited moderate cytotoxic activity across all cancer cell lines, with $IC_{50}$ values ranging from 26.30 μM (MDA-MB-231) to 54.35 μM (Caco-2). While these values are notably higher than those of doxorubicin (3.18–12.49 μM), aotaphenazine nonetheless demonstrated broad-spectrum anticancer potential, particularly in aggressive and resistant subtypes such as MDA-MB-231 (26.30 μM) and HeLa (37.79 μM). Importantly, the reduced potency relative to doxorubicin may reflect a less aggressive cytotoxic mechanism, which could translate into improved safety and reduced off-target toxicity *in vivo*.

**Table 1. *In vitro* Cytotoxicity ($IC_{50}$, μM) of aotaphenazine and doxorubicin across human cancer and normal cell lines.**

| Comp. | *In vitro* cytotoxicity $IC_{50}$ (μM) • | | | | | | | | |
|---|---|---|---|---|---|---|---|---|---|
| | WI-38 | WISH | HePG-2 | MCF-7 | HCT-116 | PC-3 | MDA-231 | Caco-2 | Hela |
| DOX | 6.72±0.5 | 8.72±0.7 | 4.50±0.2 | 4.17±0.2 | 5.23±0.3 | 8.87±0.6 | 3.18±0.1 | 12.49±1.1 | 5.57±0.4 |
| Aotaphenazine | 69.86±3.6 | 84.72±4.2 | 42.83±2.5 | 31.93±2.0 | 35.46±2.2 | 48.89±2.9 | 26.30±1.8 | 54.35±3.2 | 37.79±2.3 |

• $IC_{50}$ values are presented as the mean±SD of three replicate determinations.

In normal cell lines (WI-38 and WISH), aotaphenazine showed markedly lower cytotoxicity, with $IC_{50}$ values of 69.86 μM and 84.72 μM, respectively, compared to doxorubicin's values of 6.72 μM and 8.72 μM. This 10-fold or greater selectivity window highlights aotaphenazine's potential for differential toxicity, sparing healthy cells to a much greater extent than the non-selective cytotoxic profile of doxorubicin. This safety profile may be particularly advantageous in the context of drug resistance or dose-limiting cardiotoxicity, a known issue with anthracyclines.

The moderate cytotoxicity observed across various tumor types aligns with aotaphenazine's intercalative mode of action, which may produce cytostatic effects rather than immediate cell death. Additionally, the results suggest that aotaphenazine's antitumor efficacy may be mechanistically distinct or slower-acting than that of doxorubicin, warranting further investigation into its effects on cell cycle arrest, apoptosis induction, and Topo II-mediated DNA damage.

In summary, while aotaphenazine demonstrates lower cytotoxic potency compared to doxorubicin, it offers an encouraging therapeutic profile with broad anticancer activity and significantly reduced cytotoxicity toward normal cells. These findings support its development as a lead compound with favorable selectivity and pave the way for chemical optimization aimed at enhancing potency while retaining or improving safety.

**2.2.3. Apoptosis and cell death analysis.** To evaluate the mode of cell death induced by aotaphenazine, Annexin V-FITC/Propidium Iodide (PI) staining followed by flow cytometric analysis was performed on MDA-231 cells (Table 2 and Fig 8). The control group exhibited a predominance of viable cells (98.46%), with negligible proportions undergoing early apoptosis (0.10%), late apoptosis (1.12%), or necrosis (0.32%), confirming baseline cellular integrity in untreated conditions.

In stark contrast, treatment with aotaphenazine resulted in a dramatic shift in cell viability and death profiles. The percentage of viable cells dropped sharply to 6.08%, indicating widespread loss of membrane integrity and metabolic function. Notably, a significant proportion of cells were found in the late apoptotic phase (42.80%), accompanied by 20.98%

**Table 2. Annexin V-FITC/PI flow cytometric quantification of cell death induced by aotaphenazine.**

| Sample | Viable cells % | Early apoptosis % | Late apoptosis % | Necrosis % |
|---|---|---|---|---|
| Control | 98.50 | 0.09 | 1.13 | 0.34 |
| Aotaphenazine | 6.08 | 20.98 | 42.80 | 30.15 |

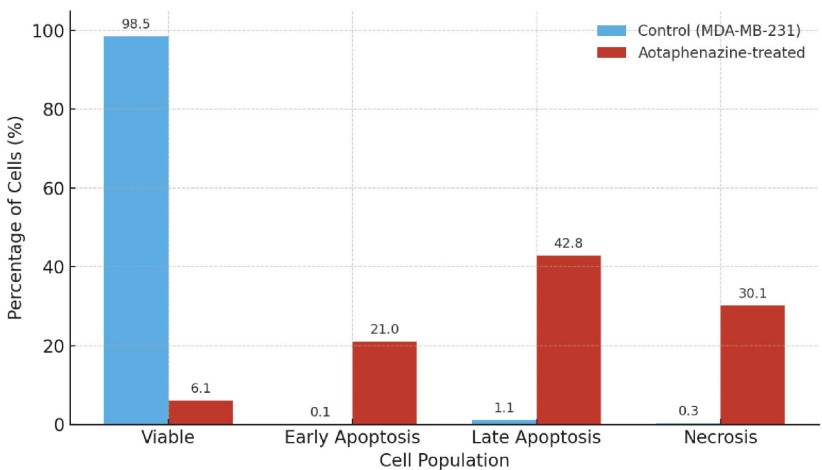

**Fig 8. Histogram for Annexin V-FITC/PI flow cytometry, showing the distribution of viable, apoptotic, and necrotic cell populations in aotaphenazine-treated MDA-MB-231 cells comparing control untreated cells.**

undergoing early apoptosis, suggesting that aotaphenazine triggers a strong apoptotic response, progressing from early to late stages of programmed cell death. Furthermore, 30.15% of the cells were necrotic, reflecting either primary necrosis or secondary necrosis following apoptosis, a phenomenon frequently observed in highly cytotoxic or late-stage apoptotic conditions.

These results clearly demonstrate that aotaphenazine induces extensive cell death predominantly through apoptosis, with a combined apoptotic population exceeding 63%, along with a notable necrotic component. This pattern is consistent with intercalating agents that disrupt DNA integrity and interfere with Topo II function, ultimately activating intrinsic apoptotic pathways. The significant late apoptosis rate also aligns with aotaphenazine's observed binding behavior in molecular simulations, supporting its role as a potent cytotoxic agent capable of inducing irreversible cell death.

Collectively, these findings establish aotaphenazine as a compound with robust pro-apoptotic activity, and reinforce its candidacy as a lead molecule for anticancer drug development, especially in tumors sensitive to apoptosis-inducing agents

**2.2.4. Cell cycle analysis.** To assess the impact of aotaphenazine on cell cycle progression in triple-negative breast cancer cells, MDA-MB-231 cells were analyzed via propidium iodide (PI) staining and flow cytometry (Table 3 and Fig 9). In the untreated control group, cells displayed a typical distribution pattern with 43.80% in G2/M phase, 16.55% in G1 phase, and 10.85% in S phase, while 43.50% fell within the sub-G1 population (P1)—which often includes debris or apoptotic fragments, as MDA-MB-231 cells tend to show basal apoptosis under stress-free conditions.

Upon treatment with aotaphenazine, a remarkable shift in cell cycle distribution was observed. The most striking change was a dramatic accumulation of cells in the S phase (43.99%), compared to just 10.85% in the control group. This nearly four-fold increase in S-phase population indicates that aotaphenazine likely induces cell cycle arrest

**Table 3. Cell cycle phase distribution in MDA-MB-231 cells following aotaphenazine treatment.**

| Sample | P1% | G1% | S % | G2% |
|---|---|---|---|---|
| Control MDA-231 | 43.50 | 16.55 | 10.85 | 43.80 |
| aotaphenazine/ MDA-231 | 29.22 | 27.84 | 43.99 | 6.66 |

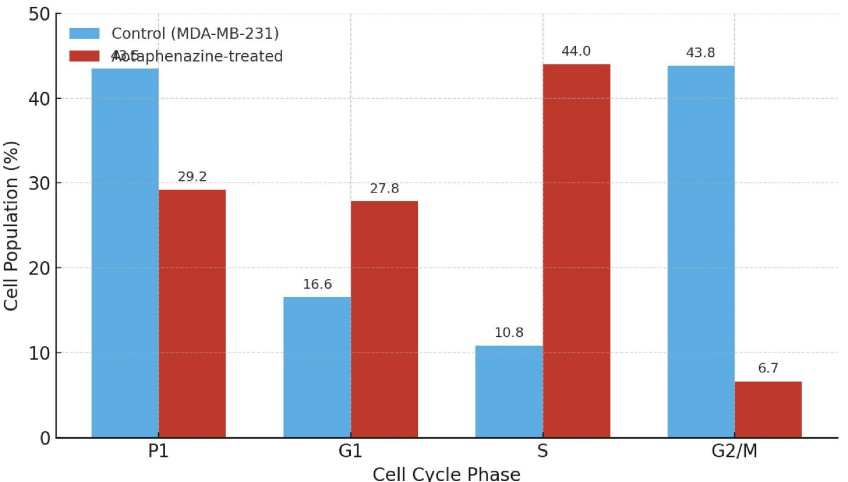

**Fig 9. Histogram showing the cell cycle phase distribution in MDA-MB-231 cells following 48 h of aotaphenazine treatment.** Compared with the control.

at the DNA synthesis stage, thereby preventing proper DNA replication and progression into G2/M. Concomitantly, the G2/M phase population dropped sharply to 6.66%, further supporting a blockade at the S-phase checkpoint. Additionally, an increased proportion of cells in G1 phase (27.84% vs. 16.55%) may reflect partial arrest at the G1/S boundary, potentially due to DNA damage or replication stress signals initiated by aotaphenazine's intercalative action.

Interestingly, the sub-G1 population (P1) decreased from 43.50% to 29.22% following treatment. While a decrease in P1 might seem paradoxical, it likely reflects aotaphenazine's shift in mechanism from basal apoptosis to active cell cycle interference, redirecting cell fate toward apoptotic cell death via replication stress rather than direct DNA fragmentation at early timepoints. This hypothesis is consistent with the high S-phase arrest and previously discussed apoptotic data showing extensive late-stage apoptosis and necrosis.

Collectively, these results suggest that aotaphenazine exerts its cytotoxic effect, at least in part, by disrupting cell cycle progression, specifically inducing S-phase arrest in MDA-MB-231 cells. This mechanism complements its known DNA intercalation and Topo II inhibitory activity, providing further support for its potential as a multifaceted anticancer agent capable of halting proliferation and inducing cell death through both cell cycle modulation and apoptosis.

## 3. Experimental

### 3.1. Molecular docking analysis

The molecular interaction between aotaphenazine and the DNA–Topoisomerase II (Topo II) complex (PDB ID: 3QX3) was systematically investigated following an optimized computational workflow established in our previous studies [31,32]. The study encompassed the following key steps: (1) Protein structure preparation — the crystal structure of the DNA–Topo II complex (PDB ID: 3QX3) was preprocessed by adding missing hydrogen atoms, assigning appropriate protonation states at physiological pH (7.4), optimizing hydrogen-bonding networks, and performing restrained energy minimization to relieve steric clashes; (2) Ligand preparation — the 3D structure of aotaphenazine was generated and subjected to conformational sampling to identify low-energy geometries, followed by partial atomic charge calculation using the Merck Molecular Force Field (MMFF94) to ensure accurate electrostatic representation; and (3) Docking grid definition — a receptor grid was centered on the ATP-binding pocket of Topo II, defined based on the co-crystallized ligand coordinates in the reference structure, to accurately capture the enzyme's active site environment for docking and subsequent interaction analysis. Complete protocols are archived in the S1 File.

### 3.2. MD simulation studies

To characterize the dynamic behavior and binding stability, we conducted a 400-ns all-atom MD simulation of the aotaphenazine–Topo II complex using GROMACS 2021 with the CHARMM36 force field. The system was solvated in a TIP3P water box (10 Å padding) with 150 mM NaCl, followed by energy minimization (steepest descent), equilibration (NVT and NPT ensembles), and production runs (2 fs timestep). Comparative trajectory analysis of apo (unliganded) versus holo (aotaphenazine-bound) Topo II revealed key conformational changes [33]. All simulation parameters and CHARMM-GUI [34] input configurations are documented in S1 File.

### 3.3. Binding free energy calculations (MM-GBSA)

Quantitative binding energetics were derived using the gmx_MMPBSA toolkit with the MM-GBSA method. From 100 equally spaced trajectory frames, we calculated: Total binding free energy, Per-component contributions (van der Waals, electrostatic, polar/nonpolar solvation) and Per-residue decomposition for amino acids within 10 Å of aotaphenazine [35]. The protocol is detailed in S1 File.

### 3.4. Protein-ligand interaction fingerprint (ProLIF) analysis

A comprehensive residue-specific interaction analysis was performed on all simulation frames using the ProLIF Python package (v1.0.0) [36]. The algorithm quantified: Hydrogen bonding patterns (distance < 3.5 Å, angle > 120°), Hydrophobic contacts (≤4.5 Å) and π-stacking/ionic interactions. Interactions with >70% occurrence frequency were classified as biologically significant. Key binding residues for aotaphenazine were identified through this statistical approach. Complete interaction matrices are provided in S1 File.

### 3.5. Protein-ligand interaction profiler (PLIP) analysis

The MD trajectory was processed through the following workflow [37]:

- **Clustering**: TTClust algorithm (RMSD cutoff = 2.0 Å) identified 3 dominant conformational states (C1-C3)

- **Representative Frame Selection**: Centroid structures from each cluster were extracted

- **Interaction Mapping**: PLIP webserver generated:

  ◦ 2D interaction diagrams (hydrogen bonds, π-effects, salt bridges)

  ◦ 3D visualization files (.pse) rendered in PyMOL (v2.5.2).

- **Quantification**: Interaction persistence (%) across clusters was calculated

  All PLIP configuration files and PyMOL session archives (S1 File).

### 3.6. Compound Sourcing

Aotaphenazine, previously isolated and characterized based on its $^1$H, $^{13}$C, 2D, as well as Ms spectroscopic data [38], was generously provided by *Prof. Masami Ishibashi* (Graduate School of Pharmaceutical Sciences, Chiba University, Japan).

### 3.7. *In vitro* topo II Inhibition assay

Aotaphenazine's inhibitory potency against human Topoisomerase II (Topo II) was quantified using an ELISA-based Topo II inhibition assay kit (Abcam, Cambridge, United Kingdom) as described before [39,40]. Briefly, serial dilutions of aotaphenazine and the reference inhibitor doxorubicin were prepared in assay buffer and incubated with the recombinant human Topo II enzyme and supercoiled plasmid DNA substrate under optimized reaction conditions. The degree of enzyme inhibition was measured colorimetrically at 450 nm, and $IC_{50}$ values were determined from dose–response curves using nonlinear regression analysis. All experiments were performed in triplicate to ensure reproducibility. Full experimental conditions and image analysis methods appear in the S1 File.

### 3.8. Cytotoxicity and antiproliferative profiling

The cytotoxic effects of aotaphenazine were assessed using the MTT colorimetric assay [41,42] to determine cell viability across a diverse panel of nine human cell lines. The panel included two normal cell lines—WI-38 (lung fibroblasts) and WISH (amnion epithelial cells)—and seven cancer cell lines, namely HepG-2 (liver carcinoma), MCF-7 and MDA-MB-231 (breast adenocarcinoma), HCT-116 and Caco-2 (colorectal carcinoma), PC-3 (prostate carcinoma), and HeLa (cervical adenocarcinoma). Cells were seeded in 96-well plates and treated with serial dilutions of aotaphenazine for 48 hours, followed by incubation with MTT reagent (0.5 mg/mL) for 4 hours. The formazan crystals were dissolved in DMSO, and absorbance was measured at 570 nm using a microplate reader. Doxorubicin served as a reference cytotoxic agent in all experiments. Cell viability was calculated relative to untreated controls, and $IC_{50}$ values were determined by nonlinear

regression analysis. Results are expressed as mean ± standard deviation (SD) of three independent biological replicates (S1 File).

### 3.9. Apoptosis and cell cycle analysis by flow cytometry

The mechanistic effects of aotaphenazine on MDA-MB-231 cells were evaluated through flow cytometric analysis using two complementary approaches: (1) cell cycle distribution was assessed via propidium iodide (PI) DNA staining, and (2) apoptosis induction was quantified through Annexin V-FITC/PI dual staining [43]. For cell cycle analysis, cells were fixed in 70% ethanol, treated with RNase A (100 µg/mL), and stained with PI (50 µg/mL) prior to acquisition. Apoptosis assays were performed on freshly harvested cells using Annexin V-FITC and PI according to the manufacturer's instructions. Flow cytometric acquisition and analysis were conducted with standard gating strategies applied to distinguish cell cycle phases ($G_0/G_1$, S, and $G_2/M$) and apoptotic subpopulations (viable, early apoptotic, late apoptotic, and necrotic) (S1 File).

## Conclusion

In this study, we report the first detailed computational and experimental investigation of aotaphenazine, a rare phenazine derivative, as a promising Topoisomerase II inhibitor and anticancer lead compound. Molecular docking and MD simulation analyses revealed that aotaphenazine intercalates stably within the DNA–Topo II complex, engaging in persistent π-stacking and hydrophobic interactions with key nucleobases. MM-GBSA calculations further confirmed its favorable binding energetics, primarily driven by van der Waals forces, consistent with a classical intercalative mechanism. Functionally, aotaphenazine demonstrated nanomolar Topo II inhibitory activity ($IC_{50}$ = 45.01 nM), comparable to doxorubicin (30.16 nM). *In vitro* cytotoxicity analysis showed moderate anticancer activity across several human cancer cell lines ($IC_{50}$ = 26.30–54.35 µM) and markedly reduced toxicity in normal WI-38 and WISH cells ($IC_{50}$ = 69.86 µM and 84.72 µM, respectively), indicating a favorable safety profile. Flow cytometry analysis further revealed early (20.98%) and late apoptosis (42.80%), S-phase cell cycle arrest (43.99%), and a significant reduction in the G2/M population in MDA-MB-231 cells, consistent with its Topo II inhibitory mechanism. Collectively, these findings establish aotaphenazine as a viable and structurally unique lead compound with potential for further development as a selective Topo II-targeting anticancer agent. Its natural origin, structural tractability, and selective cytotoxic profile underscore its promise for future medicinal chemistry optimization. Further studies, including structural analog design, *in vivo* efficacy, and toxicity profiling, are warranted to advance aotaphenazine toward clinical applicability.

## Supporting information

**S1 File. Supporting information file includes detailed molecular docking protocols; MD simulation input files and parameters; MM-GBSA workflow scripts and energy outputs; ProLIF interaction matrices; PLIP analysis and PyMOL session files; Topoisomerase II inhibition assay data; cytotoxicity assay raw readings and analyses; and flow cytometry data.**
(PDF)

## Acknowledgments

We thank Prof. Masami Ishibashi (Chiba University, Japan) for his kind provision of aotaphenazine. The authors would like to thank AlMaarefa University, Riyadh, Saudi Arabia, for supporting this research. Not applicable. This study did not involve human participants, animal subjects, or clinical data requiring ethical review. Not applicable. The research exclusively utilized computational methods and *in vitro* assays with established cell lines, eliminating the need for participant consent.

## Author contributions

**Conceptualization:** Ahmed M. Metwaly.

**Investigation:** Ahmed M. Metwaly, Wael M. Afifi, Eslam B. Elkaeed, Aisha A. Alsfouk.

**Methodology:** Mohamed S. Abdelfattah.

**Project administration:** Ahmed M. Metwaly, Ibrahim H. Eissa.

**Software:** Ibrahim H. Eissa, Eslam B. Elkaeed, Ibrahim M. Ibrahim.

**Writing – original draft:** Ahmed M. Metwaly.

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
