## [Decision Letter · Decision Letter 0]

27 Oct 2025

Dear Dr. Metwaly,

Based on the reviewers’ comments, the manuscript requires minor revision before it can be considered for publication. The experimental section should be expanded with sufficient methodological details and appropriate citations. The figure and table legends need to be more detailed and self-explanatory, and proper citations for the chemical characterization of Aotaphenazine (¹H NMR, ¹³C NMR, and MS data) should be provided. Figures 2 and 7 should be improved—specifically by including both 2D and 3D representations for DOXO and Aotaphenazine in complex with DNA-Topo II, and by clarifying images C1–C3. The cell cycle and apoptosis assay section should be revised to include control cell data for comparison with the Aotaphenazine-treated MDA-MB-231 cells. Finally, the conclusion section should be concise and clearly summarize the in vitro findings.

We look forward to receiving your revised manuscript.

Kind regards,

Abdullahi Ibrahim Uba

Academic Editor

PLOS ONE

Journal Requirements:

This research was funded by Princess Nourah bint Abdulrahman University Researchers Supporting Project number (PNURSP2025R116), Princess Nourah bint Abdulrahman University, Riyadh, Saudi Arabia

5. Please remove all personal information, ensure that the data shared are in accordance with participant consent, and re-upload a fully anonymized data set.

Reviewers' comments:

Reviewer's Responses to Questions

**Comments to the Author**

1. Is the manuscript technically sound, and do the data support the conclusions?

Reviewer #1: Yes

Reviewer #2: Yes

2. Has the statistical analysis been performed appropriately and rigorously?

Reviewer #1: Yes

Reviewer #2: Yes

3. Have the authors made all data underlying the findings in their manuscript fully available?

Reviewer #1: Yes

Reviewer #2: Yes

4. Is the manuscript presented in an intelligible fashion and written in standard English?

Reviewer #1: Yes

Reviewer #2: Yes

Reviewer #1: Dear Author,

I have gone through the manuscript, it is fine in majority of aspects. There are few aspects that needs attention like:

1. The experimental section needs to be more elaborate with proper references.

2. In the references, the authors may provide full details of the authors instead of mentioning et al.

3. Under conclusions the details of in vitro studies should be clearly discussed and it should be brief.

4. The legends to the figures and tables should be more elaborate and self explanatory.

5. Under cell cycle and apoptosis assay, only Aotaphenazine treated MDA-MB-231 cells are shown. Why the control cell data is not shown in the figure.

Reviewer #2: Reviewer Comments

Give the citation reference for the characterization of Aotaphenazine’s chemical structure, like 1HNMR/13CNMR & MS analysis.

Fig. 2: Include the two-dimensional pictures for each reference DOXO and Aotaphenazine using the DNA-Topo II complex, supported with each 3D graphic to better explain the many types of amino acid interactions.

Fig. 7: Images C1, C2, and C3 require further clarification.

**Do you want your identity to be public for this peer review?** For information about this choice, including consent withdrawal, please see our Privacy Policy

Reviewer #1: **Yes: ** Narayana Nagesh

Reviewer #2: **Yes: ** Ammar A. Razzak Mahmood

---

## [Author Response · Author response to Decision Letter 1]

5 Nov 2025

Dear editors,

Firstly, the authors would like to sincerely thank the Editor and Reviewers for their valuable time, constructive feedback, and insightful comments. Their suggestions have greatly contributed to improving the clarity, rigor, and overall quality of our manuscript. We have carefully revised the paper in accordance with all recommendations as detailed below:

Reviewer #1: Dear Author,

I have gone through the manuscript, it is fine in majority of aspects. There are few aspects that needs attention like:

1. The experimental section needs to be more elaborate with proper references.

Response: We thank the reviewer for this valuable comment. The Experimental Section has been expanded with detailed methodological descriptions and appropriate literature references.

2. In the references, the authors may provide full details of the authors instead of mentioning et al.

Response: We thank the reviewer for this helpful suggestion. The reference list has been revised to include the full author names for all citations, replacing et al. where applicable, in accordance with the journal’s referencing guidelines

3. Under conclusions the details of in vitro studies should be clearly discussed and it should be brief.

Response: We appreciate the reviewer’s helpful comment. The conclusion has been revised to briefly summarize the in vitro findings, including Topo II inhibition, cytotoxicity in cancer and normal cells, and flow cytometry results, providing a clearer and more focused summary.

4. The legends to the figures and tables should be more elaborate and self explanatory.

Response: We sincerely thank the reviewer for this valuable suggestion. All figure and table legends have been thoroughly revised to include detailed experimental descriptions, key parameters, and concise interpretations

5. Under cell cycle and apoptosis assay, only Aotaphenazine treated MDA-MB-231 cells are shown. Why the control cell data is not shown in the figure.

Response: We thank the reviewer for this important observation. In the revised version, the control (untreated) MDA-MB-231 cell data have now been included alongside the aotaphenazine-treated samples in both the cell cycle and apoptosis figures. This addition allows for a clear visual comparison between treated and control groups.

Reviewer #2: Reviewer Comments

Give the citation reference for the characterization of Aotaphenazine’s chemical structure, like 1HNMR/13CNMR & MS analysis.

Response: We thank the reviewer for this valuable suggestion. The reference detailing the full spectroscopic characterization of aotaphenazine (¹H NMR, ¹³C NMR, and MS data) has now been cited in the revised manuscript as Reference [38], which reports its isolation and structural elucidation by Prof. Masami Ishibashi and co-workers. Notably, one of the co-authors of that study, Dr. Mohamed S. Abdelfattah, who was the first author of the isolation paper, is also a co-author of the present manuscript.

Fig. 2: Include the two-dimensional pictures for each reference DOXO and Aotaphenazine using the DNA-Topo II complex, supported with each 3D graphic to better explain the many types of amino acid interactions.

Response: We thank the reviewer for this excellent suggestion. The revised Figure 2 now includes both the (2D) and (3D) interaction diagrams for doxorubicin (DOXO) and aotaphenazine bound to the DNA–Topo II complex. The updated panels clearly illustrate hydrogen bonding, π–π stacking, and hydrophobic interactions with key amino acid residues and nucleobases, providing a more comprehensive visualization of the binding modes.

Fig. 7: Images C1, C2, and C3 require further clarification.

Response: We thank the reviewer for this helpful observation. The legend and figure description for Figure 7 have been revised to provide clearer explanations of images C1, C2, and C3

---

## [Decision Letter · Decision Letter 1]

19 Nov 2025

Aotaphenazine, a Rare Hydrophenazine, Targets Topoisomerase II with Anticancer Efficacy: In Silico to In Vitro Evidence

PONE-D-25-39652R1

Dear Dr. Metwaly,

We’re pleased to inform you that your manuscript has been judged scientifically suitable for publication and will be formally accepted for publication once it meets all outstanding technical requirements.

Kind regards,

Abdullahi Ibrahim Uba

Academic Editor

PLOS ONE

Reviewers' comments:

Reviewer's Responses to Questions

**Comments to the Author**

Reviewer #1: All comments have been addressed

Reviewer #2: All comments have been addressed

2. Is the manuscript technically sound, and do the data support the conclusions?

Reviewer #1: Yes

Reviewer #2: Yes

3. Has the statistical analysis been performed appropriately and rigorously?

Reviewer #1: Yes

Reviewer #2: Yes

4. Have the authors made all data underlying the findings in their manuscript fully available?

Reviewer #1: Yes

Reviewer #2: Yes

5. Is the manuscript presented in an intelligible fashion and written in standard English?

Reviewer #1: Yes

Reviewer #2: Yes

Reviewer #1: Dear Author,

I have gone through the manuscript. Now after incoroprating the modifications suggested by the reivewer's, the manuscript looks much better. It may be accepted for publication.

Reviewer #2: (No Response)

what does this mean?

---

## [Editor Report · Acceptance letter]

PONE-D-25-39652R1

PLOS ONE

Dear Dr. Metwaly,

I'm pleased to inform you that your manuscript has been deemed suitable for publication in PLOS ONE. Congratulations! Your manuscript is now being handed over to our production team.

Kind regards,

on behalf of

Dr. Abdullahi Ibrahim Uba

Academic Editor

PLOS ONE